# AI Model for Industry Classification Based on Website Data

**Timotej Jagrič and Aljaž Herman ***

Institute of Finance and Artificial Intelligence, Faculty of Economics and Business, University of Maribor, Razlagova 14, 2000 Maribor, Slovenia; timotej.jagric@um.si
* Correspondence: aljaz.herman@um.si; Tel.: +386-2-2290-261

**Abstract:** This paper presents a broad study on the application of the BERT (Bidirectional Encoder Representations from Transformers) model for multiclass text classification, specifically focusing on categorizing business descriptions into 1 of 13 distinct industry categories. The study involved a detailed fine-tuning phase resulting in a consistent decrease in training loss, indicative of the model's learning efficacy. Subsequent validation on a separate dataset revealed the model's robust performance, with classification accuracies ranging from 83.5% to 92.6% across different industry classes. Our model showed a high overall accuracy of 88.23%, coupled with a robust F1 score of 0.88. These results highlight the model's ability to capture and utilize the nuanced features of text data pertinent to various industries. The model has the capability to harness real-time web data, thereby enabling the utilization of the latest and most up-to-date information affecting to the company's product portfolio. Based on the model's performance and its characteristics, we believe that the process of relative valuation can be drastically improved.

**Keywords:** industry classification; BERT transformer; business descriptions; multiclass text classification; AI

## 1. Introduction

The bedrock of numerous business research endeavors and economic analyses lies in industry classification schemes. These systems are pivotal for gauging economic activity, executing business censuses, pinpointing peers and competitors, defining market share, evaluating company performance against benchmarks, and crafting sector indices. The seamless execution of these tasks would be unattainable without the organizational structure offered by industry classification schemes [1].

The accurate classification of businesses is a cornerstone in various economic and financial analyses [1]. It facilitates the assessment of market competition, influences government regulatory decisions, guides capital market research, and contributes to economic studies examining investment and innovation. The accuracy of industry classification is emphasized as fundamental for guaranteeing the integrity and validity of statistical inferences drawn from empirical data [2].

Identifying and assigning companies to industries presents significant challenges. The first difficulty is the vendor-specific nature of assignments that raises concerns about consistency across data vendors. Following this, diversification complicates the affiliation of companies with detailed industries. For instance, revenue-based rules may lead to biased classifications. This issue might be addressed by allowing companies to belong to multiple industries or treating affiliation as a fuzzy problem [3].

In the paper [4] authors warn that changes in industry codes, varying data vendors, and evolving affiliations can result in Type I and Type II sampling errors. Type I occurs when a company is erroneously assigned to an industry, and Type II occurs when a company is not assigned to its actual industry. Quantifying these errors involves benchmarking assignment methods and comparing alternatives.

In [1] authors mention that these big industry classification schemes (ICS), such as NACE or GICS, must find a delicate stability between granularity and practicality. The upkeep of extensive company universes, encompassing business lines, products, and services with ICS assignments, demands substantial investments in both resources and manpower from business information providers and government entities. The progress and sustainability of ICSs will advance in tandem with the perceived return on investment by ICS producers and the discerned value that customers and users derive from these classification systems.

In the paper [5], the author continues that activity-based classification systems, like SIC and NAICS, have notable limitations, primarily stemming from their inability to accurately reflect the structure of industries. Despite efforts to align with industry structures, such systems still fall short. They fail to depict the interconnectedness between firms in adjacent categories, leading to oversights in understanding the relationships within sectors. Additionally, such systems can result in inconsistent classifications for similar firms based on how they operate rather than their core purpose. Furthermore, as industries evolve with technological advancements like microsystems and software, these systems struggle to keep pace, leading to inaccuracies in classification. For instance, the rise in complex products reliant on diverse components challenges traditional categorization methods. Moreover, innovations like nanotechnology further complicate industry classifications, highlighting the ongoing struggle of activity-based systems to adapt to changing economic landscapes.

To address the limitations of these activity-based classification systems, we have used an industry classification model that offers significant advantages. While traditional systems struggle to capture the intricacies of evolving industries, our model offers a dynamic solution. Although initially trained on a specific dataset, its adaptable nature allows for easy adjustments and fine-tuning to accommodate various industry structures and changes over time for the optimal business classification.

With the exponential growth of digital information, automated classification has become not just advantageous, but essential [6]. This paper presents the innovative application of BERT, a Large Language Model (LLM) in the realm of business classification based on its description.

The BERT model has become a cornerstone in the area of NLP (natural language processing) for various tasks such as classification, named entity recognition, and question answering [7]. BERT's advantage is using transformer's bidirectional training, because on the contrary, the primary constraint of conventional language models lies in their unidirectional nature. This constraint significantly restricts the options available for architectural choices during the pre-training process. These models typically employ a left-to-right architecture, wherein each token possesses the ability to attend solely to the tokens that precede it in the self-attention layers of the transformer [8]. On the basis of [9], we can claim that bidirectional models can understand the meaning of the text in more detail than models that are trained in a single direction. The transformer encoder comprehends the complete sequence of words simultaneously. While often described as bidirectional, a more precise characterization would be non-directional. This unique attribute empowers the model to grasp the contextual nuances of a word by considering its entire surroundings, encompassing both the left and right contexts [8].

In this paper, we present a comprehensive overview of our model's architecture, highlighting how the BERT framework has been fine-tuned for the task of business classification. This phase is critical in harnessing the full potential of BERT [10], ensuring that the model is not just powerful in general language understanding, but also acutely tuned to the subtleties and specifics of business categorization. In the end, we will present our results and conclusions, demonstrating the efficacy of using BERT for business classification.

Our model's architecture is based explicitly on the BERT transformer and was built on almost 70,000 business descriptions. Its simple and flexible architecture allows for rapid adaptation to disaggregation. Moreover, our model's capability to harness real-time web data positions it as a powerful tool for industry practitioners. By enabling the utilization of

the latest and most up-to-date information affecting a company's product portfolio, our approach offers practical implications for decision makers seeking dynamic and accurate insights in real-world business scenarios. A simplified integration scheme is shown in Figure 1. One of the advantages is also that the model is considered to be small among large language models, which allows it to be used locally, and with that comes increased privacy and reduced dependence on external servers, ensuring that sensitive data stay within the user's control. This will be supported by a comprehensive description of the model's structure and parameters presented in the subsequent sections.

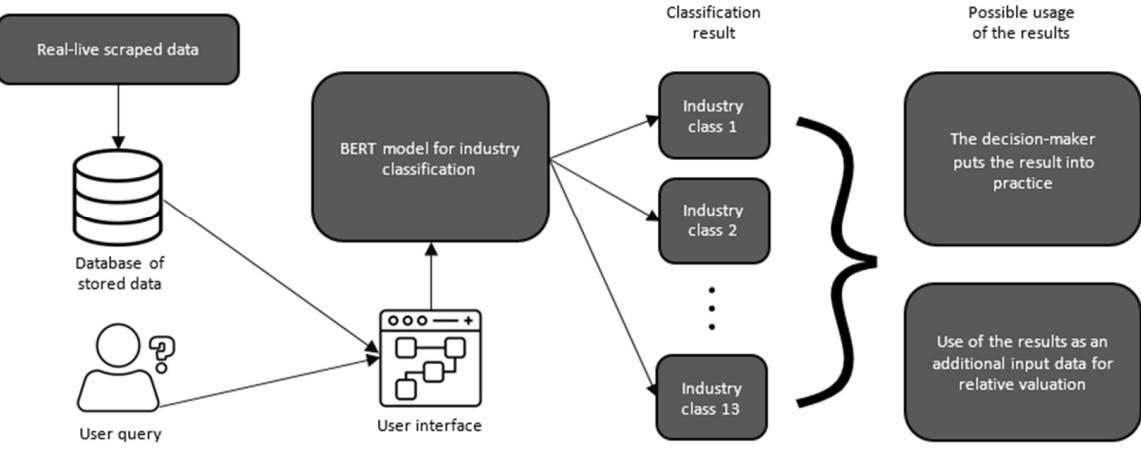

**Figure 1.** Integrated solution scheme.

## 2. Related Work

In a related project [11] conducted on the same dataset [12], the researcher employed a series of sophisticated data processing and machine learning techniques to categorize businesses into corresponding industries. The primary methodology involved a combination of text preprocessing, vectorization, and clustering analysis to derive meaningful categorizations from the web-scraped data. The core of the project was the application of a cluster analysis. The researcher employed a TensorFlow-based implementation of K-means clustering, optimized for GPU processing, to categorize the businesses into distinct clusters. The ideal number of clusters was identified using both the Elbow Method and Silhouette Scores, with the final decision being to create twelve clusters. They were found to represent distinct business categories, each with their own unique textual footprint within the data.

In the quest for more accurate and insightful industry classification, researchers have explored a variety of models, each designed to capture distinct facets of the dynamic business environment. In their study [13], authors showcased the application of a Naive Bayes (NB) classifier on a dataset focused on the industry sector. This dataset comprised company web pages, systematically categorized within a hierarchical structure of 71 industry sectors. The reported findings revealed that the multinomial NB classifier achieved an accuracy level of up to 0.74. Additionally, the multivariate Bernoulli model demonstrated an accuracy rate of up to 0.46.

In [14] authors advocate for the utilization of Support Vector Machines (SVM) employing a one-vs-all strategy and error-correcting output coding in the context of the industry sector dataset. Their findings indicate a noteworthy improvement in performance compared to Naive Bayes (NB).

In his paper [15] author illustrates the efficacy of k-Nearest Neighbors (kNN) and the DragPushing strategy-based kNN classifier (DPSKNN) methodologies on a subset comprising 48 sectors from the industry sector dataset. The micro-F1 score for the kNN classifier is reported as 0.8188, with a corresponding macro-F1 score of 0.8235. Conversely, DPSKNN exhibits a slightly superior performance, achieving a micro-F1 score of 0.8544 and a macro-F1 score of 0.8585.

In more recent studies [16], authors employed four distinct models to analyze company descriptions. Firstly, they implemented a linear model—a perceptron without hidden layers. The perceptron's input was the sum of unigram vectors. The second approach introduced a customized linear model, incorporating unigrams and bigrams as features while still using one-hot vector representations. The third model, GloVe, aimed to integrate context vectors by employing 300-dimensional GloVe vectors instead of one-hot representations. Lastly, they tested the ULMfit algorithm, a sophisticated classification model utilizing context vectors. In contrast to the previous models, ULMfit leveraged a custom language model based on AWD-LSTM for direct context vector generation, offering a more intricate and flexible approach to language modeling and classification training. Regarding the micro-F1 scores, the first three models showed a performance with a score just over 0.92, and GloVe at 0.906. On the other hand, the macro-F1 scores indicated that hot-bigram outperformed the others with a score of 0.712, followed by hot-unigram and GloVe with a score that was just under 0.69, and ULMfit at 0.641.

In [17] authors developed a model for automated industry classification using deep learning methods. They introduced a multilayer perceptron that has the capability to acquire knowledge from imperfect labels while also incorporating and blending verified examples as they become accessible.

Several related studies have been identified where both the model and the purpose were similar to ours. In their work [18] authors focused on companies listed on the Chinese National Equities Exchange and Quotations (NEEQ). Their dataset covered 17,604 annual business reports, which limited the use of the model, as well as its durability. Authors [19] used companies shown in Shanghai Stock Exchange and Oriental Fortune Net as a dataset upon which they built a model that was based on two main architectures—a convolutional neural network (CNN) and the BERT transformer. Similarly, authors [20] combined a CNN and bidirectional long short-term memory networks in addition to using the BERT transformer. On the contrary to these, our dataset includes almost 70,000 different business descriptions, is well-generalized, and is based solely on the architecture of the BERT transformer.

## 3. Materials and Methods

In the context of using a BERT transformer model for multiclass industry classification based on a business description, the phase of data collection assumes significance. This stage entails the acquisition of a representative and diverse dataset that is reflective of the intended classification task. In our case, we needed a dataset with two crucial entities, a business description text of their initial purpose and corresponding labels, which, in this case, are industry categories. In this paper, we used a comprehensive database [12] containing a substantial volume of data with 73,974 rows and 10 columns, but for the needs of this research, we only kept 2 of them—"Meta Description", which contains short business description of the work, and "Category", in which the corresponding industry class is attributed. Although our model was initially trained on a pre-existing dataset, its adaptability and efficacy are demonstrated by its seamless integration with live data.

Equally critical is a comprehensive data inspection process, which encompasses data exploration, preprocessing, and cleansing—identifying and correcting errors, inconsistencies, and inaccuracies in a dataset. In her work [21] author addresses issues such as missing values, data imbalances, and the removal of irrelevant or duplicate entries. Data analysis tools offer substantial utility in the business context, but their effectiveness hinges upon the meticulous cleansing of data before yielding meaningful outcomes. Otherwise, the entire data processing pipeline succumbs to the axiom of "garbage in, garbage out", rendering the results far less valuable than anticipated by the business teams. It is imperative to acknowledge that real-world data are inherently imperfect, given the inevitability of errors that can manifest in intricate and unforeseeable ways.

Following these processes, a total of 7088 rows were systematically removed due to the presence of missing or inaccurate values, ensuring the data's integrity and quality. As

a result, the new refined database now consists of 66,886 rows, signifying a considerable reduction from its initial size of 73,974 rows. This streamlined database also underwent transformations to enhance its consistency and usability. Specifically, special characters, when present, were excluded from the data, further enhancing the quality of the dataset and ensuring that it adheres to standardized formatting practices. These data cleansing and transformation steps were essential in preparing the database for reliable and meaningful analysis [22]. An example of the transformed dataset is shown in Table 1.

**Table 1.** Example of the dataset (source: adopted from [12]).

| Business Description | Industry Class |
|---|---|
| American Association of Neuromuscular & Electrodiagnostic Medicine-AANEM is dedicated to advancing neuromuscular and electrodiagnostic medicine by providing physician education, advocacy efforts, and resources for patients with muscle and nerve disorders. | Healthcare |
| Searching for a private security and investigation company in Midland, TX? Look no further than Finley Investigations & Security, Inc. If you are in need of security guards for construction and oil field sites, or need a private investigator, we are the ones to call. Visit our site today. #sep#PROVIDING QUALITY SECURITY GUARDS, CAMERA MONITORING, AND PRIVATE INVESTIGATION SERVICES IN THE PERMIAN BASIN | Commercial Services & Supplies |
| Aluminum & Zinc Ingots—Custom alloyed to your exact specifications. From aerospace to automotive to industrial, we manufacture & distribute the widest array of recycled and primary casting alloys in the USA. | Materials |

Our labeling column encompasses a total of 13 distinct categories. Each category represents a unique sector or domain, offering a structured and systematic way to classify and analyze the content within the dataset based on business descriptions. The names of the categories and the corresponding number of enterprises in that category are shown in Table 2.

**Table 2.** Table of categories and corresponding number of enterprises.

| Category Name | Label | Number of Cases | Training Set | Testing Set |
|---|---|---|---|---|
| Commercial Services & Supplies | 0 | 5856 | 4685 | 1171 |
| Healthcare | 1 | 6534 | 5227 | 1307 |
| Materials | 2 | 2418 | 1934 | 484 |
| Financials | 3 | 6278 | 5022 | 1256 |
| Energy & Utilities | 4 | 5162 | 4130 | 1032 |
| Professional Services | 5 | 6655 | 5324 | 1331 |
| Corporate Services | 6 | 6442 | 5153 | 1289 |
| Media, Marketing & Sales | 7 | 5798 | 4638 | 1160 |
| Information Technology | 8 | 5426 | 4341 | 1085 |
| Consumer Discretionary | 9 | 2611 | 2089 | 522 |
| Industrials | 10 | 3073 | 2458 | 615 |
| Transportation & Logistics | 11 | 5727 | 4582 | 1145 |
| Consumer Staples | 12 | 4906 | 3925 | 981 |

As authors [23] mentioned, it is important to properly encode labels for a better operation of the BERT model. According to this, we encoded the names of industry classes as consecutive numbers, starting with 0. The encoded labels are presented in Table 2.

By reserving a distinct portion of the dataset for testing, the train–test split offers a rigorous evaluation of the model's performance on previously unseen data, highlighting its

effectiveness and robustness. Additionally, it aids in detecting issues like overfitting [24]. Regarding this, we had to split our data into two subsets—training and validation sets. We followed Pareto's principle [25] and used one of the common ratios [24] for splitting the data, 80:20, which means that 80% (53,508) of the data were used for training the model and the rest of them (13,378) were used to confirm that the model was properly built, meaning that the outcomes on the training set were not the result of overfitting, rather, the model was properly trained and is able to classify companies into industry classes accurately. At this point, we also had to pay attention to the distribution of classes within the dataset. Class distribution refers to the proportion of each class (category or label) in the dataset. In many real-world scenarios, datasets are imbalanced, meaning that some classes may be significantly underrepresented compared to others. Ignoring class distribution during data splitting can lead to a range of issues in model training and evaluation [26]. When class distribution is not adequately considered, there is a risk of having an insufficient number of instances from one or more classes in the training set. This scenario can severely hinder the model's ability to learn from those underrepresented classes. The model may struggle to make accurate predictions for these classes, as it lacks sufficient examples to understand their patterns and characteristics. Conversely, in the absence of consideration for class distribution, certain classes may not be represented in the testing set. In this scenario, the model's performance on these classes cannot be evaluated, as there are no instances available for testing. This lack of representation can result in misleading performance metrics and hinder the model's overall effectiveness [26].

So, we took into consideration the risk and maintained the distribution by class to prevent an inaccurate model performance. The distribution by class is shown in Table 2.

In the remainder of this section, we will present the method we used. For interested readers, we suggest the following sources [10,27,28] for more detailed information on the description of the method and clarification of its features.

In the paper [8] authors explained, that, to ensure that BERT can effectively handle a wide range of downstream tasks, it is equipped with a flexible input representation that can unambiguously accommodate both single sentences and pairs of sentences (such as question–answer pairs) within a single token sequence. Here, the term "sentence" is not limited to traditional linguistic sentences; rather, it can represent any contiguous span of text. A "sequence" in the context of BERT pertains to the token sequence used as an input, which may encompass either a single sentence or two sentences combined. For tokenization, which is the process of breaking down text into smaller units called tokens, BERT employs WordPiece embeddings [29] with a vocabulary size of 30,000 tokens. For our needs, the base model with a 30,000-word vocabulary is adequate, striking a balance between computational efficiency and effective text representation. In every input sequence, the initial token is always a distinct classification token [CLS]. The final hidden state corresponding to this [CLS] token serves as the comprehensive representation of the entire sequence, particularly valuable for classification tasks. In cases where sentence pairs are involved, they are combined into a single sequence. To distinguish between these sentences, two strategies are employed. First, they are separated by a special [SEP] token. Second, segment embedding is added to each token, indicating whether it belongs to sentence A or sentence B. But, in the case of classifying business descriptions into industry classes, the use of special [SEP] tokens and segment embeddings is not necessary, especially because the sentences in the business description are closely related and form a coherent text.

Authors continue that the BERT framework encompasses two key steps: pre-training and fine-tuning. In the pre-training phase, the model undergoes training using vast amounts of unlabeled data while engaging in various pre-training tasks. Following this, in the fine-tuning step, the BERT model is initially set up with the pre-trained parameters, serving as a foundation. It then proceeds to fine-tune all of its parameters using labeled data specific to downstream tasks. Notably, for each individual downstream task, distinct models are fine-tuned. Even though they share the same set of pre-trained parameters, the

fine-tuning process tailors these models to the distinct requirements of their individual tasks [8].

Originally proposed by authors [7], the transformer architecture serves as the backbone for BERT. It replaces recurrent layers with self-attention mechanisms and feed-forward neural networks (FFNN), enabling parallel computation and efficient learning. BERT utilizes the transformer architecture, which is based on an attention mechanism capable of understanding the contextual relationships between words or sub-words within a given text. This transformer architecture comprises two core components: an encoder, responsible for processing the input text, and a decoder, which generates predictions for specific tasks. However, BERT's primary objective is to create a language model, and for this purpose, it exclusively relies on the encoder module, which is presented in Figure 2.

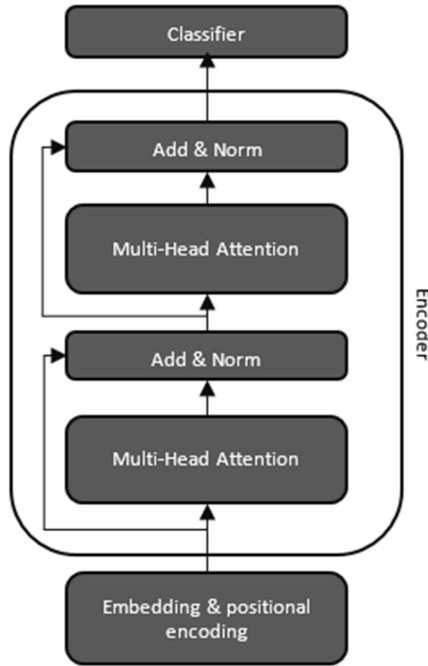

**Figure 2.** Transformer architecture (source: adopted from [7]).

The encoder component within the transformer architecture is structured as a stack of $N = 6$ identical layers, each comprising two sub-layers. The first sub-layer incorporates a multi-head self-attention mechanism, while the second sub-layer features a straightforward, position-wise fully connected feed-forward network. To enhance the flow of information, a residual connection, following the principles of residual networks [30], surrounds each of these two sub-layers. This is coupled with layer normalization [31], which ensures that the output of each sub-layer conforms to a standard. To enable the implementation of these residual connections, all sub-layers in the model, in addition to the embedding layers, generate outputs with the same dimensionality $d_{model}$. This consistent dimensionality facilitates the flow of information and the fusion of various sub-components within the encoder [7].

When performing multiclass classification with BERT, the labels should typically be encoded as integers [23]. Each integer represents the class label or category to which a specific text belongs. The choice of encoding may vary, but it is essential that the labels are discrete integers, with each integer corresponding to a unique class. According to the explanation given, we encoded the names of categories into suitable forms, meaning we assigned consecutive numbers to each category starting with 0.

After the data were properly preprocessed, we used tokenization, which is a pivotal preprocessing step for BERT-based models [8]. We used a tokenization provided by the Hugging Face transformers library. We set the parameter "return attention mask" as true,

which means that the tokenizer will generate attention masks for the input sequences. Attention masks help the model to focus on actual input tokens while ignoring padding tokens. With the parameter "pad to max length" set as true, we ensured that all sequences in the batch (subset) were padded to the same maximum length, which we set to 256. This choice balances the need for comprehensive context representation with practical considerations, as longer sequences would significantly increase the computational demands, potentially exceeding the available resources for model training and inference. The parameter "return tensors to pt" specifies the format of the output. Setting it to "pt" indicates that the output should be in the PyTorch tensors format, allowing us to train the model on graphical processing units (GPU) [32]. A detailed explanation on the importance of this concept will be presented subsequently in the paper.

In the BERT transformer model, the word embedding step converts each token ID into a high-dimensional vector [7]. These vectors serve as the initial representations of the words and are designed to capture the semantic meaning of each token [33]. This embedding is typically performed through a lookup table, where each unique token ID is mapped to a predefined vector in the embedding space. These vectors are trainable parameters, meaning they are updated during the backpropagation process to better capture the semantics of words in the context of the specific task. The embedded vectors are the first representations that are fed into the subsequent layers of the transformer model. This is a crucial step, as it transforms discrete tokens into a form that the model can understand and manipulate [8].

Positional encoding is added to the word embeddings to give the model information about the position of each token in the sequence. Unlike traditional RNNs (Recurrent Neural Networks) or LSTMs (Long Short-Term Memory), transformers do not have a built-in sense of sequence order, so positional encoding is crucial for tasks that depend on the order of words. The positional encoding is usually generated using a specific mathematical formula involving sine and cosine functions. The formula generates a unique encoding for each position that is then added to the corresponding word embedding vector. Mathematically, the positional encoding for position $p$ in dimension $d$ is calculated as:

$$PE(p,d) = \sin\left(\frac{p}{10,000^{\frac{2d}{d_{model}}}}\right) \text{ if } d \text{ is even,}$$

$$PE(p,d) = \cos\left(\frac{p}{10,000^{\frac{2d}{d_{model}}}}\right) \text{ if } d \text{ is odd.}$$

The resulting positional encodings have the same dimension as the word embeddings, so they can be added together. After generating these positional encodings, they are added element-wise to the word embedding vectors [34], which is shown in Figure 3. The resulting vectors, which now contain both semantic and positional information, are what are passed into the subsequent layers of the transformer model [7].

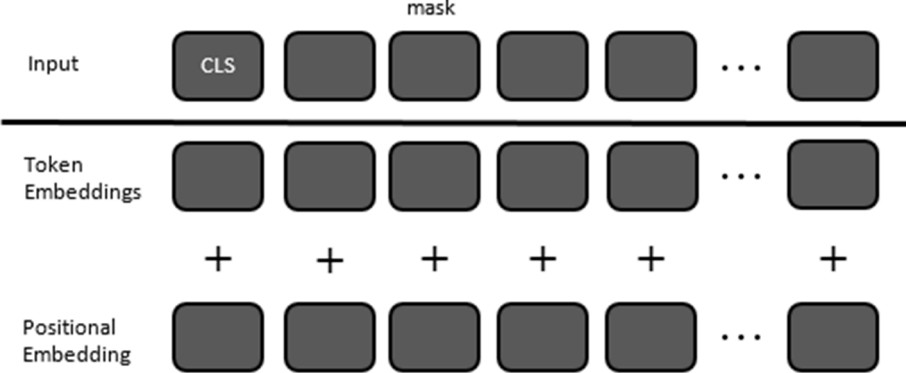

**Figure 3.** Input representation (source: adopted from [8]).

Next in the transformer's architecture is an attention function [7], which serves as a way to process information. It is like a map that takes in a query along with an assortment of input key–value pairs and provides an output. In this context, query, keys, values, and output are represented as vectors. The output, in particular, is determined by calculating a weighted sum of the values. These weights are derived from a compatibility function that measures how well the query aligns with each key. Essentially, the compatibility function helps to decide how much importance or attention to give to each value, and the weighted sum of these values forms the final output.

Regarding BERT's multi-head self-attention, it is essential to begin by understanding the concept of scaled dot-product attention, which can be defined as follows:

$$Attention(Q, K, V) = softmax\left(\frac{QK^T}{\sqrt{d_k}}\right)V,$$

where $Q$ represents the matrix containing queries, $K$ is the matrix holding the keys, $V$ is the matrix storing the values, and $d_k$ represents the dimension shared by the $Q$ and $K$ matrices [9].

Now, we can introduce the multi-head attention. With multi-head attention, the model gains the ability to collectively focus on information residing in various representation subspaces and at distinct positions within the input data. It is defined as:

$$MultiHead(Q, K, V) = Concat(head_1, \dots, head_h)W^O,$$

where $head_i = Attention\left(QW_i^Q, KW_i^K, VW_i^V\right)$, projections are the parameter matrices $W_i^Q \in \mathbb{R}^{d_{model} \times d_k}, W_i^K \in \mathbb{R}^{d_{model} \times d_k}, W_i^V \in \mathbb{R}^{d_{model} \times d_v}$ and $d_v$ is the dimension of the values.

Multi-head attention involves a process where the queries, keys, and values are projected multiple times using distinct learned linear transformations. These projections aim to map them to new dimensions such as $d_k$ for queries and keys and $d_v$ for values. Once these different versions of the queries, keys, and values are obtained, the attention function is applied independently to each set. This generates output values, each with a dimension of $d_v$. These outputs are then brought together by concatenation and further projection to create the final values. The term "self-attention" signifies that all the queries, keys, and values originate from the same source or context. This self-attention mechanism allows the model to understand the relationships and dependencies between different elements within the same input sequence [7].

Residual connections, also known as skip connections, allow the input of a layer to be added to its output, which can help to mitigate the exploding and vanishing gradient problem during training [35]. This is particularly important in deep networks like BERT.

Let us denote the input to a layer (after any previous layer normalizations and operations) as $X$. If $F(X)$ represents the operation performed by the layer (such as multi-head attention or feed-forward neural network), the output of the layer with the residual connection is:

$$Y = F(X) + X.$$

This equation means that the output of the layer $Y$ is the sum of the original input $X$ and the processed input $F(X)$. The dimensions of $X$ and $F(X)$ must be the same to make this element-wise addition possible [36].

After the residual connection, layer normalization [31] is applied. This operation normalizes the data across the features for each data point in a batch, which helps to stabilize the learning process.

Let us consider $Y$ as the output from the residual connection. The layer normalization process involves calculating the mean and variance for each data point across the features:

$$\mu = \frac{1}{H}\sum_{i=1}^{H} y_i \text{ and } \sigma^2 = \frac{1}{H}\sum_{i=1}^{H}(y_i - \mu)^2,$$

where $H$ is the number of features (the dimensionality of the hidden layer) and $y_i$ represents the individual features of the output $Y$.

The next step is normalizing the output $Y$ using the mean $\mu$ and variance $\sigma^2$:

$$\hat{y}_i = \frac{y_i - \mu}{\sqrt{\sigma^2 + \epsilon}},$$

where $\epsilon$ is a small constant for numerical stability.

In the end, we have to apply learnable parameters $\gamma$ (scale) and $\beta$ (shift) to the normalized output.

$$LN(y_i) = \gamma\hat{y}_i + \beta,$$

where $LN(y_i)$ is the final output. We can label it as $LN$.

A feed-forward neural network [7] is the second sub-layer in the model's architecture. It comes after layer normalization and is applied to each position separately and identically. This means that the same feed-forward neural network is applied to each position, but it operates independently on the inputs from each position. The purpose of this sub-layer is to introduce non-linearity into the model, which allows the network to learn more complex patterns.

The FFNN operates on each position's vector independently but in the same way. This is different from the self-attention mechanism that considers other positions. It also introduces non-linear capabilities to the model, enabling it to learn complex patterns beyond what linear transformations can capture.

The FFNN structure consists of two linear transformations with non-linearity between them. The input to the first linear transformation is the output from the layer normalization. Let us denote this input as $LN$, where this is a matrix with rows corresponding to positions in the sequence and columns corresponding to the model's hidden units. The first linear transformation applies weights $W_1$ to the input $LN$ and adds a bias term $b_1$. This transformation projects the input into a higher-dimensional space.

Then, the non-linear activation function *ReLU* [37] is applied:

$$H = ReLU(LN \cdot W_1 + b_1),$$

where $H$ is the output of the first transformation. The *ReLU* activation function is defined as:

$$ReLU(z) = max(0, z),$$

where $z$ represents each element of the matrix $H$.

Finally, the output of the ReLU activation is passed through a second linear transformation with its own weights $W_2$ and bias $b_2$. This transformation projects the data back to the original dimensionality of the model's hidden layers as:

$$F = HW_2 + b_2,$$

where $F$ is the final output of the FFNN, $W_2$ is the second weight matrix, and $b_2$ is the second bias vector [7].

After the FFNN, residual connection that adds the input of the FFNN sub-layer to its output is typically applied, followed by a layer normalization. This, like before, helps to mitigate the risk of exploding and vanishing gradients [35] and allows the model to learn identity functions, which is beneficial for deep networks [7].

The pre-training method closely adheres to established practices in language model pre-training, drawn from a substantial pre-training corpus that includes the BooksCorpus, which comprises 800 million words [38], and the English Wikipedia, consisting of 2500 million words. When using Wikipedia, text passages were specifically extracted and lists, tables, and headers were excluded. In paper [8] authors also point out that a noteworthy aspect of this approach is the use of a document-level corpus as opposed to a

shuffled sentence-level corpus like the Billion Word Benchmark [39]. This choice enables the extraction of extended, coherent sequences of text.

For pre-training, we used the BERT model for sequence classification using the BERT-base-uncased pre-trained model as its foundation. We set the "num labels" parameter to match the number of classes in our classification task, dynamically adjusting the model's output dimension to accommodate the classification needs. Additionally, we can control whether the model should provide attention weights and hidden states as part of its output. In this instance, both output attentions and output hidden states were set to false, indicating that the model would not return attention weights or hidden states by default. These settings are often preferred for standard sequence classification tasks such as ours [32].

The pre-training of BERT commences by immersing it in two distinct unsupervised tasks—masked language modeling (MLM) and next sentence prediction (NSP).

Fine-tuning in the BERT model is a straightforward process due to the versatile self-attention mechanism of the transformer architecture, which will be explained in the following sections of this paper. This self-attention mechanism allows BERT to effectively model a diverse range of downstream tasks, irrespective of whether they involve processing single texts or text pairs. This flexibility is achieved by simply adjusting the inputs and outputs as needed for each task. For tasks that involve text pairs [40,41], a common approach is to separately encode both texts and then perform bidirectional cross-attention. However, BERT takes a different approach by utilizing the self-attention mechanism to seamlessly merge these two stages. When encoding a concatenated text pair, the self-attention mechanism inherently includes bidirectional cross-attention between the two sentences. To fine-tune BERT for specific tasks, we customize the inputs and outputs to match the requirements of each task. For input, the sentences A and B from the pre-training phase can be analogous to different types of pairs, such as sentence pairs in paraphrasing, hypothesis–premise pairs in entailment, question–passage pairs in question answering, or even a single text paired with an empty context for text classification or sequence tagging. At the output, token representations are fed into an output layer for token-level tasks like sequence tagging or question answering. Meanwhile, the [CLS] representation is directed to an output layer for classification tasks such as entailment or a sentiment analysis. Compared to the resource-intensive pre-training phase, fine-tuning is a relatively efficient process, making it cost-effective and practical for various NLP applications [8].

For handling the training data efficiently, we set up data loaders [42], which play a vital role in the training process, as they load and manage batches of input data for efficient model training. We used a function called DataLoader, which is provided by deep learning frameworks like PyTorch. The critical part here is the sampler parameter, which we set to random. This means the data loader will randomly shuffle the training data before creating batches. Shuffling is essential to prevent the model from learning patterns based on the order of the data, ensuring that they generalize well [43]. Lastly, the batch size parameter specifies how many data samples are processed together in each batch [44]. The choice of batch size is a crucial hyper parameter that influences the training process and memory usage. Larger batch sizes can speed up training but require more memory. We decided to set up the batch size to 3. This choice optimally balances the utilization of available resources, mitigating memory constraints while concurrently enhancing the model's capacity to learn from diverse instances within each batch, thus fostering an improved generalization performance.

Training extensive deep neural networks on massive datasets causes significant computational challenges. In response to this, there has been a notable upswing in the exploration of large-batch stochastic optimization techniques as a means to address this computational difficulty [45]. In our case, we used AdamW [46], which originates from a stochastic optimizer called Adam [47], which is an efficient stochastic optimization technique that operates with minimal memory usage and relies solely on first-order gradients. Adam calculates personalized learning rates for various model parameters based on estimations

of the gradients' first and second moments. The term itself is derived from its core principle, which involves adaptive moment estimation. Adam has gained widespread popularity for training deep neural networks because of its reduced need for hyper parameter tuning and its outstanding performance [46]. To enhance the model's ability to generalize, Adam is often used alongside a squared $l_2$ regularizer, known as Adam-$l_2$. However, even more superior results can be achieved by employing AdamW, a variant that effectively separates the gradient of the regularizer from the update rule of Adam-$l_2$. This decoupling mechanism contributes to improved training outcomes [48].

We set the learning rate, a parameter that determines the step size that the optimizer takes during each iteration of training, to $10^{-5}$ and the eps to $10^{-8}$, a parameter that defines a small constant value used to prevent division by zero when computing the adaptive learning rates in the optimizer to ensure numerical stability [49].

We also used a predefined framework learning rate scheduler called get linear schedule with warmup [50]. Its primary purpose is to manage the learning rate throughout the training process. The name linear signifies that this scheduler linearly adjusts the learning rate during training. It commences with an initial learning rate and progressively decreases it over a specified number of training steps or epochs. The warmup aspect indicates an initial warmup phase. During this phase, the learning rate gradually escalates from a very small value to the initial learning rate. This warmup phase is valuable for ensuring a stable training process and preventing early convergence issues. The scheduler's core objective is to strike a balance between the stability and convergence speed. To employ this scheduler, we had to specify the optimizer [32]. We used AdamW, as was already described.

The last step in our model was to build a training loop [32]. We also defined a procedure or, more precisely, an evaluation metric [36] for monitoring outcomes. It is instrumental in several aspects of model evaluation. Firstly, it underscores the class-specific performance, highlighting how the model fares with each individual class. This is particularly vital in scenarios where the importance of classes varies, ensuring that critical categories are adequately addressed. Secondly, it adeptly handles class imbalance. In datasets where some classes are overrepresented, a general accuracy metric might be misleading. This method, however, offers a more detailed and accurate representation of the model's performance across all classes. Additionally, it plays a crucial role in identifying weaknesses. By pinpointing classes where the model's performance is lacking, it guides targeted improvements in both the model and data collection strategies.

## 4. Results

In this segment, we present an in-depth analysis of the results obtained from the training and validation phases of our BERT-based model. The training phase results provide insights into the model's learning process over epochs, as evidenced by changes in training loss. Following this, we delve into the performance of the model on the validation set, which serves as a crucial indicator of its generalization capabilities and real-world applicability. This allows for a complete understanding of the model's effectiveness in classifying business descriptions into predefined industry categories.

The results of the training process, shown in Table 3, span over five epochs, showing a consistent decrease in loss, indicating an improvement in the model's ability to classify the data accurately. The choice of fine-tuning the model over five epochs is optimal, as it achieves a balance between leveraging pre-existing knowledge from the pre-trained model and adapting to the task-specific data, ensuring effective classification.

This downward trend in training loss is a positive indicator of the model's learning efficiency. The initial loss of 0.46 in the first epoch, which is relatively high, suggests that the model began with limited knowledge about the data's structure and the classification task. However, as the training progressed, the model rapidly improved, with the loss decreasing by approximately 28% from epoch 1 to epoch 2, and by about 30% from epoch 2 to epoch 3. The rate of decrease in loss slowed down in the subsequent epochs, with a reduction of 26% from epoch 3 to epoch 4, and a further 12% decrease from epoch 4 to epoch 5.

**Table 3.** Training loss for each epoch.

| Epoch | Training Loss |
|-------|---------------|
| 1 | 0.46 |
| 2 | 0.33 |
| 3 | 0.23 |
| 4 | 0.17 |
| 5 | 0.15 |

This pattern of rapid improvement in the initial epochs followed by a slower rate of improvement in later epochs aligns with observations made in other studies utilizing deep learning models for text classification [8,10]. The initial steep decline in loss can be attributed to the model quickly learning major patterns in the data, while the gradual decrease in later epochs indicates the model's refinement in understanding and classifying more nuanced aspects of the data.

The final training loss of 0.15 at epoch 5 demonstrates the model's effective adaptation to the task, suggesting a high level of accuracy in classifying business descriptions into the correct industry categories. However, it is important to note that training loss alone is not a comprehensive indicator of model performance. Evaluation metrics such as accuracy on a validation set are essential to fully understand the model's effectiveness [51].

In conclusion, the training process of the BERT-based model for this multiclass text classification task shows promising results, with a consistent decrease in training loss across epochs. This indicates a successful learning trajectory, aligning with trends observed in similar applications of deep learning models in text classification tasks.

At this point, it is also worth noting the significant impact of utilizing GPUs (Graphics Processing Units) for accelerating our language model training on such a vast database. Unlike CPUs (Central Processing Units), GPUs boast essential parallel processing capabilities, revolutionizing the execution of complex neural network operations and markedly enhancing the efficiency of the entire training process. When our model was executed on a CPU, the training process consumed over 40 h per epoch. This extended timeframe is primarily attributed to the sequential nature of CPU processing. However, upon transitioning to GPU acceleration, we observed a transformative reduction in the training duration, achieving an astonishingly swift 6 h per epoch. The key advantage lies in the parallel architecture of GPUs, which enables the simultaneous execution of multiple operations. Neural network computations, inherently parallelizable, experience a substantial speedup when processed on GPUs. This not only expedites model training, but also unlocks the potential for handling larger datasets and more complex architectures.

The analysis of the validation set results for our BERT-based model reveals significant insights into its performance across different industry classes, which is shown in Table 4. The model demonstrates a commendable level of accuracy in classifying business descriptions, with accuracies ranging from 83.5% to 92.6% across the 13 industry categories. Such variation in accuracy across different classes is a common observation in multiclass classification tasks and can be attributed to factors like class imbalance, the varying complexity of class-specific features, and the amount of training data available per class [52,53].

The model achieved a notably high accuracy in classes like healthcare (92.6%), consumer staples (91.4%), and financials (90.7%) This could indicate that the descriptions in these categories have distinct features that the model learns effectively, leading to more accurate predictions.

All the other industry classes exhibited accuracies between 83.5% and 89%, with the category of corporate services showing an accuracy just above 89%. This suggests a generally robust model performance across a diverse set of classes.

**Table 4.** Confusion matrix of distributions of predicted values for each industry class with number of total cases and accuracy.

| True (Class)/ Predicted (Label) | 0 | 1 | 2 | 3 | 4 | 5 | 6 | 7 | 8 | 9 | 10 | 11 | 12 | Total Cases | Accuracy |
|---|---|---|---|---|---|---|---|---|---|---|---|---|---|---|---|
| Commercial Services & Supplies | **1010** | 6 | 16 | 5 | 25 | 10 | 17 | 19 | 17 | 18 | 10 | 11 | 7 | 1171 | 86.3% |
| Healthcare | 10 | **1210** | 5 | 3 | 4 | 14 | 16 | 7 | 15 | 10 | 1 | 9 | 3 | 1307 | 92.6% |
| Materials | 18 | 2 | **412** | 1 | 9 | 1 | 0 | 1 | 5 | 0 | 23 | 6 | 6 | 484 | 85.1% |
| Financials | 11 | 10 | 3 | **1139** | 8 | 30 | 15 | 4 | 16 | 8 | 3 | 7 | 2 | 1256 | 90.7% |
| Energy & Utilities | 40 | 3 | 7 | 9 | **916** | 4 | 2 | 7 | 8 | 2 | 16 | 8 | 10 | 1032 | 88.8% |
| Professional Services | 18 | 11 | 1 | 36 | 6 | **1165** | 30 | 13 | 28 | 8 | 3 | 6 | 6 | 1331 | 87.5% |
| Corporate Services | 16 | 20 | 1 | 10 | 10 | 23 | **1148** | 20 | 13 | 8 | 3 | 4 | 13 | 1289 | 89.1% |
| Media, Marketing & Sales | 6 | 4 | 2 | 10 | 4 | 22 | 14 | **1028** | 39 | 6 | 5 | 9 | 11 | 1160 | 88.6% |
| Information Technology | 25 | 22 | 3 | 19 | 11 | 31 | 14 | 18 | **908** | 6 | 5 | 10 | 13 | 1085 | 83.7% |
| Consumer Discretionary | 13 | 5 | 3 | 6 | 2 | 7 | 13 | 12 | 6 | **436** | 6 | 9 | 4 | 522 | 83.5% |
| Industrials | 14 | 3 | 21 | 5 | 15 | 3 | 2 | 2 | 8 | 2 | **519** | 14 | 7 | 615 | 84.4% |
| Transportation & Logistics | 24 | 8 | 5 | 3 | 10 | 5 | 18 | 7 | 9 | 7 | 28 | **1015** | 6 | 1145 | 88.6% |
| Consumer Staples | 8 | 6 | 2 | 2 | 3 | 6 | 27 | 6 | 3 | 4 | 9 | 8 | **897** | 981 | 91.4% |

The distribution of accurate predictions across classes was relatively balanced, with no class showing an extremely low accuracy. This balance is crucial in multiclass classification tasks to ensure that the model does not favor certain classes over others, a challenge often addressed in the literature [54].

The variation in accuracies across classes may also reflect the inherent complexity and distinctiveness of the textual data in each category. Classes with a lower accuracy, such as consumer discretionary (83.5%) and information technology (83.7%), might own more ambiguous or overlapping features with other classes, making classification more challenging.

To sum up, the validation results demonstrate the model's effective learning and generalization capabilities across a range of industry classes. The high accuracy in certain classes suggests that the model is particularly adept at capturing and utilizing distinctive features in those categories. Meanwhile, the moderate to high accuracy in other classes indicates a well-rounded performance.

Furthermore, shown in Table 4, the confusion matrix offers a detailed view of the model's performance in classifying business descriptions into their respective industry classes [55]. The matrix shows the distribution of predicted values against the true values for each industry class, allowing for a nuanced analysis of the model's classification accuracy and its potential areas of confusion.

The high values along the diagonal of the matrix, representing correct classifications, indicate strong performances in all classes.

Besides the overall accuracy, which is 88.23%, we calculated some additional metrics to obtain even more in-depth insights into our model's performance.

The precision per class, which shows us the proportion of correctly classified instances among the instances classified as that class, ranged from 0.83 for the commercial services & supplies category to 0.92 for the healthcare category. These values signify the accuracy of the positive predictions made by the classifier for each specific class. A precision of 0.83 implies that approximately 83% of the instances predicted as belonging to the commercial services & supplies category were indeed classified correctly, while a precision of 0.92 indicates a higher accuracy level, with around 92% of instances in the healthcare category being correctly classified.

Next was recall, which tells us for each class the proportion of correctly classified instances among all the instances that truly belong to that class. Notably, recall ranged from 0.84 for the information technology, consumer discretionary, and industrials categories to 0.93 for the healthcare category. These findings underscore the effectiveness of the classifier in correctly identifying the instances belonging to each specific class. A recall score of 0.84

indicates that approximately 84% of instances in the mentioned categories were accurately classified by the model, while a higher recall of 0.93 for the healthcare category signifies that around 93% of instances within this class were correctly identified.

In conjunction with both, the parallel nature of these metrics underscores a balanced performance, where high precision values indicate low false-positive rates, complemented by elevated recall values signifying thorough positive instance capture, collectively contributing to a robust evaluation of the classification model across various classes.

Additionally, we used a slightly more complex metric, which is defined on the basis of the eigenvalues of the matrix [56]:

$$M_{SVD}(P) = \frac{\sum_i^N \sqrt{\lambda_i\left(\widetilde{P}'\widetilde{P}\right)}}{N},$$

where $P$ is the relative matrix of our confusion matrix, the mobility matrix ($\widetilde{P}$) is defined as $\widetilde{P} = P - I$, $I$ is an identity matrix of the same size as $P$, $\widetilde{P}'$ is the transposed matrix of $\widetilde{P}$, and $\lambda_i$ represents the eigenvalues of the matrix product $\widetilde{P}'\widetilde{P}$.

The background here are the transition matrices [57], where we use these measures to see how strong the transition is from the main (diagonal) classes to the rest. We obtained the result of 0.123, which indicates a high degree of closure of the main classes, i.e., there is relatively little leakage from them.

As it goes for the wrongfully predicted cases, there are noticeable instances of misclassification, where descriptions from one class were predicted as another [58]. This suggests certain similarities in the textual features between these classes that the model may be conflating. In the materials industry, we can see there are two industries (commercial services & supplies and industrials) which have visible deviations among others. The same observation goes for the industry of financials, where professional services havethe highest deviation. Significant deviation happens also in energy & utilities, where the most misclassified cases are from commercial services & supplies. In the category of professional services, the industries of financials, corporate services, and information technologies are the ones with the highest deviation. In media, marketing & sales the most misclassified cases fell into information technology and, in this industry, most cases fell into professional services. In industrials industry, the highest deviation is observed for materials, and in transportation & logistics, this was industrials and commercial services & supplies. In consumer staples, the highest deviation happened in corporate services. Nevertheless, the errors are relatively well-distributed across different classes, without any single class being predominantly misclassified as another. This indicates balanced learning by the model, without significant biases towards certain classes [59].

## 5. Discussion and Conclusions

This study explored the application of the BERT (Bidirectional Encoder Representations from Transformers) model for the task of multiclass text classification, focusing on categorizing business descriptions into 13 distinct industry classes. The results from both the training and validation phases demonstrate the model's proficiency in understanding and classifying complex text data.

During the training phase, the model exhibited a consistent decrease in loss across five epochs, indicating effective learning and adaptation to the task. The final training loss achieved was significantly lower than the initial loss, underscoring the model's ability to capture the fundamental attributes of the dataset.

In the validation phase, the model's performance was robust across various industry classes, with accuracies ranging from 83.5% to 92.6%. This variation in class-specific performance highlights the challenges and complexities inherent in multiclass text classification tasks. The high accuracy in certain classes suggests that the model was particularly effective

at identifying the unique characteristics of those industries. Meanwhile, the moderate to high accuracy in other classes indicates a well-rounded ability to generalize across diverse types of text data.

In comparison to various models employed for industry classification, our developed model demonstrated an accuracy of 88.23% and an F1 score of 0.88. This surpasses the performance of a Naive Bayes (NB) classifier, achieving an accuracy level of up to 0.74 [13]. The exploration of the k-Nearest Neighbors (kNN) and DragPushing strategy-based kNN classifier (DPSKNN) methodologies, while showcasing competitive micro-F1 and macro-F1 scores around of 0.82 (kNN) and 0.85 (DPSKNN) [15], did not surpass our model's accuracy. In a more recent context [16], authors employed four models. While slightly below the micro-F1 scores of some models (just over 0.92), our model notably outperforms them in terms of macro-F1 scores, where the highest macro-F1 score reaches 0.712.

The study's findings reinforce the potential of transformer-based models like BERT in handling intricate classification tasks in natural language processing. The ability of BERT to understand context and nuances in text makes it a powerful tool for business applications, where the accurate categorization of textual data can provide significant insights and operational advantages.

Despite the initial training on a pre-existing dataset, the model's capability to adapt and perform effectively is evident in its smooth integration with real-time or live data. This suggests the model's ability to apply its learned knowledge to new and dynamic information, showcasing its versatility and practical utility.

With the application of this model, we successfully demonstrated the viability of employing the BERT transformer in its small version as an LLM in classification task. Our findings indicate that, with meticulous fine-tuning, the BERT transformer can be effectively utilized for analogous tasks. This underscores the model's adaptability and opens up avenues for its application across diverse classification scenarios.

While our current implementation of BERT-based classification has yielded promising results in identifying and categorizing firms into sectors, there exist opportunities for further refinement and enhancement. One potential avenue for improvement lies in the adoption of a more sophisticated classification scheme within the BERT model. By incorporating a nuanced learning process that corrects misclassifications and applies varying degrees of penalization based on the occurrence of firms in incorrect or related sectors, we can potentially boost the model's accuracy and robustness. Additionally, adjusting the penalization strategy based on the severity of misclassifications may contribute to a more adaptive and fine-tuned classification system. A notable limitation of our current BERT-based classification model is its reliance on manually sourced business descriptions from the web. This process not only limits the volume and diversity of data, but also introduces potential biases. A promising direction for future research would be to automate the acquisition of business descriptions, utilizing advanced web scraping and natural language processing techniques. Additionally, the model's current proficiency is primarily in English, which restricts its global applicability. Addressing this limitation could involve expanding the training to encompass multilingual datasets, thereby enhancing the model's relevance in international markets. Another significant area for development is the integration of this model into a larger analytical framework. This could take the form of a hybrid system, where the BERT classifier functions as a component in conjunction with other tools like market trend analyzers or financial performance modules, offering a more comprehensive relative valuation. Lastly, the model could benefit from an expanded and more nuanced sector categorization. This would allow for finer distinctions between industries and a more accurate reflection of the current business landscape, especially in rapidly evolving or newly emerging sectors. These enhancements and expansions are crucial for advancing the model's accuracy, applicability, and utility in the dynamic field of business classification.

**Author Contributions:** Conceptualization, T.J. and A.H.; methodology, T.J.; software, A.H.; validation, T.J. and A.H.; formal analysis, T.J. and A.H.; investigation, T.J. and A.H.; resources, T.J. and A.H.; data curation, T.J. and A.H.; writing—original draft preparation, A.H.; writing—review and editing, T.J. and A.H.; visualization, A.H.; supervision, T.J.; project administration, T.J. All authors have read and agreed to the published version of the manuscript.

**Funding:** This research received no external funding.

**Institutional Review Board Statement:** Not applicable.

**Informed Consent Statement:** Not applicable.

**Data Availability Statement:** Original data were gathered from Kaggle platform by Charan Puvvala—Company Classification. 2019 at https://www.kaggle.com/datasets/charanpuvvala/company-classification (accessed on 3 October 2023).

**Conflicts of Interest:** The authors declare that they have no known competing financial interests or personal relationships that could have appeared to influence the work reported in this paper.

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
