# Peer review of "AI Model for Industry Classification Based on Website Data"

_information, doi:10.3390/info15020089_

Round 1

Reviewer 1 Report

Comments and Suggestions for Authors

General overview:

It was with great anticipation that I started reading the article titled “AI Model for Industry Classification Based on Live Data”. The text is well-organized, moving from the model architecture to the training process and then to the results.

1-The explanation of the Layer Normalization and Feed-Forward Neural Network (FFNN) components is detailed. The use of technical terms without clear definitions might hinder understanding for readers not familiar with neural network architectures.

2-The training process is well-documented, highlighting the use of BERT and the specific pre-training corpus.

GPU acceleration is mentioned but lacks a thorough explanation.

3-The presentation of results is clear, with a detailed breakdown of accuracy across different industry classes.

The discussion of misclassifications in the confusion matrix adds depth to the analysis. Moreover, the comparison with previous models adds credibility to the study. However, the lack of specific references or direct citations could be a drawback.

Ways to improve the manuscript:

Overall, the article demonstrates a good understanding of the subject and presents detailed technical information. However, some improvements in clarity and accessibility, as well as a more in-depth discussion of limitations, could enhance the overall quality of the article. Therefore:

1-The paper could benefit from a more comprehensive introduction to make it accessible to a broader audience. Clearer definitions of technical terms and concepts would enhance readability.

2-The model's limitations and potential areas for future research are not discussed (although it is mentioned some avenues for improvement), limiting the paper's completeness. I recommend making future research suggestions clearer.

Comments on the Quality of English Language

Minor editing of English language required.

Author Response

Dear Sir/Madam, 

we have written our comments in the attached file.

Best regards

Reviewer 2 Report

Comments and Suggestions for Authors

This paper is about the conceptual task-oriented fine-tuning of large language models.

The case is the business classification. LLMs might be quite broad, and these days, researchers are also looking for ways to subdivide the broadness of LLM.

I highly recommend a better re-highlighting of the contributions at the end part of the introduction by summarily paraphrasing.

On page 4, the example of the dataset samples are required. 

"The return tensors to pt specifies the format of the output." -> implicit information. 

The authors do not clearly identify their contribution and the "proposed" method in Section 2, which is very critical. 

The ideas are given like brainstorming, and the logical flow is required to be reconsidered, especially on pages 6-8. The conventional attention mechanisms and some background are mixed with the proposal. Also, these pages are too verbal, and there is no technical attitude. Flowcharts, network details (the ones including the proposed scheme), and algorithms are required to support the authors' contribution. 

Pages 9-11 hold experiment-related sharing; therefore, I am not sure why the information is presented in this section. (E.g., "For handling the training data efficiently, we set up data loaders [39], which play a vital role in the training process.")

From the initial section, the information authors claim: "One of the advantages is also that the model is considered small among large language models, which allows it to be used locally and with that comes increased privacy and reduced dependence on external servers, ensuring sensitive data stays within the user's control." should be supported by experimental section.

It is good to present the confusion matrix (CM), but there are many metrics to discuss when using the CM, so the results need to be extended. I highly recommend considering other works for comparison purposes. The authors should also better discuss the generalization and targeted (reduced) generalization behavior of their LLM model.

Language issue:

it's -> it is

Author Response

(The authors gave the same response as above.)

Reviewer 3 Report

Comments and Suggestions for Authors

In this manuscript, the authors employ the BERT model for multi-class industry classification using a kaggle dataset containing companies' textual descriptions. The topic is interesting and worth investigating.

My remarks are as follows:

1. In "Abstract" and "Introduction" sections, the authors should present their contributions in the area of text-based classification, particularly focusing on theoretical aspects.

2. The related literature part of “Introduscion” could be reorganized in a new “Related Work” section. At the end of this new section, a summary of the current trends in the applications of NLP models for industry classification should be included. The results could be summarized in tabular form.

3. The content of “Materials and Methods” section could be presented more consisely. Its current style is narrative. Additionally, the size of this section appears disproportionately larger than that of others. Acctually, GICS identifies 11 industry sectors. Why is there an application of a 13-industry sectors classification? This requires clarification.

4. The discussion part is too brief. Here, a comparion with results obtained from similar previous studies should be added including using several datasets and NPL classification methods.

5. In “Conclusions” section, study limitations should be addressed.

6. Table 3 is derived from Table 4; hence, Table 4 could be omitted.

7. The title requires revision, for example: “AI model for industry classification based on live data” -> “AI model for industry classification based on real-life data”.

8. A link to the authors’ programming code could be added.

Author Response

(The authors gave the same response as above.)

Round 2

Reviewer 1 Report

Comments and Suggestions for Authors

The authors made the necessary changes.

Author Response

Dear reviewer,

Regards

Reviewer 2 Report

Comments and Suggestions for Authors

I thank the authors for their revision.

Nevertheless, the paper requires a more comprehensive results section, as I highlighted in my previous comment (Comment 8). The provided confusion matrix offers the potential to extract a wealth of information and metrics, leading to a more quantitative discussion. The revisions appear to be more of a minor nature, and I suggest that the authors dedicate additional effort to enhancing the results section. The current version of the paper (results) does not significantly differ from a conference paper. Therefore, I recommend one more iteration, particularly focusing on presenting thorough performance metrics.

Author Response

Dear reviewer,

Regards

Reviewer 3 Report

Comments and Suggestions for Authors

Despite some improvements in the manuscript, it is still not ready for publication. My remarks are as follows:

1. The theoretical contributions are missing.

2. The meaning of "live data" is unclear. As the authors employ scraped datasets, their solution does not work in real-time. Instead, it seems more aligned with offline or batch processing. In this regard:

   First, the title should be changed; for example, "AI model for industry classification based on real-live data" -> "AI model for industry classification based on real-life data."

   Second, the authors should check their method description and clarify its characteristics.

Author Response

Dear reviewer,

Regards

Round 3

Reviewer 2 Report

Comments and Suggestions for Authors

I thank the authors for their revision.